# Total Tear IgE Levels Correlate with Allergenic and Irritating Environmental Exposures in Individuals with Dry Eye

**DOI:** 10.3390/jcm8101627

**Published:** 2019-10-04

**Authors:** Harrison Dermer, Despoina Theotoka, Charity J. Lee, Priyanka Chhadva, Abigail S. Hackam, Anat Galor, Naresh Kumar

**Affiliations:** 1Miller School of Medicine, University of Miami, Miami, FL 33136, USA; hid7@med.miami.edu (H.D.); charity.lee513@gmail.com (C.J.L.); 2Bascom Palmer Eye Institute, University of Miami, Miami, FL 33136, USA; dxt412@miami.edu (D.T.); ahackam@med.miami.edu (A.S.H.); 3Department of Ophthalmology, Illinois Eye and Ear Infirmary, University of Illinois, Chicago, IL 60612, USA; pchhadva7@gmail.com; 4Department of Public Health Sciences, University of Miami, Miami, FL 33136, USA; nkumar@med.miami.edu

**Keywords:** IgE, allergies, environment, pet dander, smoking

## Abstract

Dry eye (DE) and allergic conjunctivitis may present similarly, and it remains unclear whether some individuals have an underlying allergic component to their DE. To better understand this relationship, we performed a cross-sectional study in 75 individuals with DE symptoms and/or signs. Immunoglobulin E (IgE) levels in tear samples were quantified and home environmental exposures assessed via standardized survey. Tears were collected by Schirmer strip, and total tear IgE levels were quantified using enzyme-linked immunosorbent assay (ELISA). Data were analyzed using descriptive statistics and linear and logistic regressions. The main outcome measures were total tear IgE levels and their association with environmental exposures. The mean age of the subjects was 66.2 ± 7.8 years. Sixty-two individuals had dry eye symptoms (Dry Eye Questionnaire-5 ≥ 6), and 75 had one or more signs of DE. Detectable total tear IgE levels were observed in 76% of subjects, and 17.3% had high levels (>1 ng/mL). Individuals with exposure to pet(s) (odds ratio (OR) 11.5, *p* = 0.002) and smoke (OR 38.6, *p* = 0.008) at home were more likely to have high IgE levels compared to those not exposed. Individuals with tears collected during spring or summer were 3.9 times (*p* = 0.028) more likely to have high IgE compared to those sampled at other times of year. Subjects born in the US were 3.45 times (*p* = 0.010) more likely to have high IgE compared to individuals born outside the US. To conclude, a majority of individuals with DE symptoms and/or signs had detectable IgE levels in their tears. High tear IgE levels were correlated with allergy season and exposures in the home linked with allergy.

## 1. Introduction

Dry eye (DE) is a multifactorial condition broadly defined by inflammatory and/or neurosensory changes at the ocular surface, accompanied by derangements of the tear-film [1]. Affected individuals present with symptoms of ocular irritation, dryness, discomfort, or foreign body sensation [2] Many risk factors have been identified in DE, including internal factors such as thyroid disease, and diabetes, and external factors such as medications (e.g., diuretics, antihistamines), surgery, and environmental exposures [3]. Dry eye impacts more than 16 million Americans [4], and up to 30% of people over fifty years of age worldwide [5], and negatively impacts quality of life [3,6]. Allergic Conjunctivitis (AC) is a hypersensitivity disorder that is considered distinct from DE [7,8,9]. However, the two entities have overlapping features that include shared symptoms (discomfort and itching [10]) and signs (bulbar hyperemia and ocular surface inflammation [11,12,13,14]). Given this overlap, a question remains whether some individuals diagnosed with DE have an underlying sub-clinical component of allergy [15]. Data to support this hypothesis include the reported seasonal variation of DE, with increased symptom severity in spring, correlating with maximal pollen levels [16,17]. Furthermore, environmental conditions and exposures known to impact AC [18,19], including humidity, dust mites, and tobacco smoke have also been shown to impact DE [20,21]. Determining which factors contribute to DE in an individual patient is crucial for the delivery of precision medicine. As such, an objective marker is needed to identify when DE has an allergic component. 

Immunoglobulin E (IgE) is an antibody that plays a critical role in type 1 hypersensitivity reactions and has been used as a biomarker to assist in the diagnosis of allergic disease [22,23]. Total and allergen-specific tear IgE levels have been studied in relation to AC [24], but not in relation to DE. To bridge this knowledge gap, we investigated the frequency of detectable tear IgE levels in individuals with DE symptoms and/or signs but without a diagnosis of AC. Furthermore, we evaluated which factors were associated with high total tear IgE levels, focusing on environmental conditions and exposures associated with allergy. We hypothesized that allergenic or irritating exposures in the home would associate with high IgE levels in our cohort.

## 2. Experimental Section

Study Population: Male and female subjects were prospectively recruited from the Miami VA Medical Center eye clinic between October 2010 and December 2011. Written informed consent was obtained from all subjects after verbal explanation of the nature and possible consequences of the study.

Inclusion criteria included the presence of DE symptoms and/or signs, grossly normal eyelid and corneal anatomy, no evidence of ocular surface pathology beyond DE (e.g., corneal edema, pterygium), and no characteristic findings of AC (e.g., chemosis, conjunctival fibrosis). Based on review of the medical record, patients were excluded from participation if they had ocular or systemic conditions that could confound DE, including contact lens wear, a history of refractive surgery, use of ocular medications except for artificial tears, active external ocular pathology, cataract surgery in the last 6 months, or a history of glaucoma or retinal surgery. Systemically, individuals with human immunodeficiency virus, sarcoidosis, Sjögren’s, graft-versus host disease, collagen vascular disease, or primary immunodeficiency with elevated serum IgE (e.g., Job syndrome, Wiskott–Aldrich syndrome) were excluded. Subjects with a history of omalizumab use, symptomatic AC, or specific allergic sensitization confirmed by skin or conjunctival provocation test or serum-specific IgE were also excluded.

The Miami VA institutional review board approved the examination of patients for this study, which was conducted in accordance with the principles of the Declaration of Helsinki (IRB Protocol #3011.02) and complied with the requirements of the United States Health Insurance Portability and Accountability Act.

Clinical assessment and home environment survey: Each individual filled out a standardized questionnaire regarding DE symptoms (Dry Eye Questionnaire-5, DEQ5: score 0–22 [25]) and an additional survey of exposures associated with allergy (e.g., pets, carpets) or the aggravation of allergy (tobacco smoke) (Appendix A). All individuals underwent an ocular surface exam, which included, in the order performed: tear osmolarity (TearLab Osmolarity System, TearLab, San Diego, CA, USA) (once in each eye), tear evaporation measured as tear break up time (TBUT) (5µL fluorescein instilled in the superior conjunctiva, time measured in seconds until the first black spot appeared in the tear film, two measurements taken with 5 s blink interval between measurements and averaged), assessment of corneal epithelial cell disruption measured by staining with fluorescein (Bron scale, 0–4 scale [26]), and assessment of tear production via Schirmer’s test with anesthesia (measured as millimeters of wetting after five minutes). Sterile Schirmer test strips were utilized (TearFlo, HUB Pharmaceuticals, Plymouth, MI, USA). Corneal contact was avoided in order to limit reflex lacrimal secretion, and the strips were immediately stored in –80 °C for the follow up analysis of total tear IgE.

IgE Analysis: Schirmer strips from the right eye of each patient were used. The Schirmer strips were defrosted and total tear proteins were eluted from the wetted area of the strip into 30 µL BSS by first vortexing for 10 s, followed by gentle shaking for 1 h at room temperature, followed by an additional vortexing for 10 s. The sample was centrifuged for 1 min, and the solution was removed and kept on ice. An additional 30 µL BSS was added to the strips, and the samples were processed as above. The eluates were pooled and then stored at −80°C until analysis.

ELISA was performed on the eluted tear samples using Human IgE ELISAPRO kit (MABTECH INC West Street Cincinnati, OH, USA), following the manufacturer’s directions. Each eluted sample was diluted to a final volume of 200 µL using BSS, and equivalent volumes of 100 µL of each sample were tested in duplicate. These precautionary volume adjustments were performed to ensure standardization of tear fluid volumes in the samples. Negative controls (assay background control and blanks) were included to measure background signals from the plate and the positive control was a standard IgE stock solution provided with the kit. The limit of total tear IgE detection was 0.069 ng/mL. The ELISA was quantified using a microplate reader at 450 nm (BMG Labtech Omega microplate reader). The data were fit to a 4-parameter standard curve, and the average of each sample was calculated after subtracting the blank from the standard for each run (blank corrected values). We chose not to measure allergen-specific tear IgE because potential allergens were unknown at the time of analysis.

Statistical Analysis: All statistical analyses were performed using STATA Version 14 (STATA Inc., 2014). Descriptive analyses were employed to examine study variables. We examined the contribution of independent variables on total tear IgE, both as a continuous variable using linear regression and as an ordinal variable using ordinal logistic regression (function ologit) and treating IgE level below the detection limit as the base. For the categorical analysis, we binned our subjects into 3 groups: <0.69 ng/mL (non-detectable (ND)), ≤1 ng/mL (Low), and >1 ng/mL (High). We chose 1 ng/mL (0.417 IU/mL) as the cutoff between Low and High IgE status based on prior studies [27,28,29].

## 3. Results

### 3.1. Study Population

A total of 75 individuals with DE symptoms and/or signs, but without a diagnosis of AC were included. Mean age was 66.2 years ± 7.7, and the majority of subjects were male (87%), white (60%), and non-Hispanic (67%). Sixty-two individuals had dry eye symptoms (DEQ5 ≥ 6), and 75 had one or more signs of DE. Mean DE symptoms (via DEQ5) were in the moderate range (11.04 ± 5.32), and most individuals had evaporative DE (TBUT ≤ 8 s, 54.7%) with adequate tear production (Schirmer score ≥ 8, 91.9%). Fifty-seven individuals (76%) had detectable IgE levels in their tears (≥0.069 ng/mL) and 13 (17.3%) had High total tear IgE, defined as a value >1 ng/mL based on prior literature [27,28,29] (Table 1).

### 3.2. DE and Total Tear IgE

Total tear IgE levels were not found to correlate with DE measures, both when considering symptoms and signs of disease. For example, High tear IgE was observed in 17% of individuals with negligible DE symptoms (DEQ5 < 6), 27% of individuals with mild-moderate symptoms (DEQ5 ≥ 6 and <12), and 11% of individuals with severe symptoms (DEQ5 ≥ 12) (Table 2). While no significant correlations between total tear IgE levels and DE signs were observed, there was a positive trend between total tear IgE and tear production. Patients with ND IgE had the lowest mean Schirmer scores (9.9 ± 1.3 mm/5 min, 95% confidence interval (CI) 8.6–11.2); those with Low IgE had higher scores (12.4 ± 1.0 mm/5 min, 95% CI 11.4–13.4); and those with High IgE had the highest scores (13.6 ± 1.7 mm/5 min, 95% CI 12.9–15.3) (Table 2).

### 3.3. Total Tear IgE and Home Exposures

The presence of environmental allergens and/or irritants inside the home positively associated with detectable total tear IgE levels (Table 1). Specifically, individuals with pet(s) at home had a higher frequency of High IgE (38.5% vs. 7.0%) compared to those with no pet(s) (*p* = 0.003). Likewise, individuals who smoked (60% vs. 14.3%, *p* = 0.027) or were exposed to second hand smoke at home (60% vs. 15.6%, *p* = 0.015) had a higher frequency of High IgE compared to individuals not exposed to smoke at home. Finally, tear collection during local allergy season (spring and summer) associated with a higher frequency of High IgE (29.0% vs. 9.1%) and lower frequency of ND IgE (12.9% vs. 31.8%) compared to samples collected during fall and winter (*p* = 0.032) (Table 1). These findings were confirmed via multivariable analyses, in which total tear IgE (as a continuous variable) and total tear IgE categories (ND, Low, and High) were examined separately, adjusting for covariables including age, birthplace, smoking status, DE measures, and pet(s) (Table 3). Smoking status (OR = 38.57, *p* = 0.008) and pet(s) in home (OR = 11.54, *p* = 0.002) showed significant association both with total tear IgE at a log scale and total tear IgE categories. Measurement during spring or summer (local peak allergy seasons) (OR = 3.94, *p* = 0.028) was a significant risk factor, while birthplace outside the US was a significant protective factor (OR = 0.29, *p* = 0.010) based on categorical analyses. In the analysis of total tear IgE as a continuous variable, patients below the limit of detection were excluded, restricting the analysis to 52 patients. IgE levels among smokers and patients with a pet at home were 2.86 (*p* = 0.033) and 2.16 (*p* = 0.049) times higher than among non-smokers and patients without a pet, respectively. Even exposure to second-hand smoke had a significant positive association with total tear IgE levels *(p* < 0.01).

## 4. Discussion

We found that a majority of individuals with clinical features of DE but no diagnosis of AC had detectable total tear IgE levels. Seventeen percent had High levels, defined as >1 ng/mL. Individuals with High tear IgE levels were more likely to be exposed to allergens and irritants in their home environment. Our data provide indirect evidence that some individuals with symptoms and/or signs of DE have an allergic component underlying their disease. However, we did not find that one specific manifestation of DE associated with tear IgE, as levels were distributed equally amount the different DE measures. The only trend noted was between Schirmer score and IgE levels, with high tear production positively associating with total tear IgE. Interestingly, elevated tear production has been well described in AC [30].

Overall, the mean total tear IgE level (for those with detectable levels) in our population was 0.363 IU/mL (0.873 ng/mL), which is higher than prior reports from healthy controls (0.058 IU/mL; 95% CI 0.012–0.287), but lower than what has been reported in patients with AC (5.259 IU/mL; 95% CI 0.053–523.219) [24,31,32,33]. This is not surprising, as none of our subjects had a diagnosis of AC, but points to a possible subclinical component of allergy in a portion of our patients. However, it is important to remember that elevations in total serum and/or tear IgE are not specific for allergy, so we cannot say definitively that undiagnosed allergy was behind the tear IgE elevations we observed. A range of parasitic, helminthic, and viral infections, tuberculosis, eosinophilic granulomatosis with polyangitis, certain lymphomas and myelomas, various autoimmune diseases, and bone marrow transplantation may all result in increased total IgE in serum and perhaps in tears. Although we excluded individuals with known systemic confounders of allergy, it is possible that some individuals had undiagnosed disease.

There is biologic plausibility for the observed relationships between our noted exposures and ocular surface parameters. Pet dander is already known to drive several IgE-mediated conditions, including AC [32], so it is not surprising that exposure posed an increased risk of having high total tear IgE. Increased total tear IgE levels in spring and summer are likely due to greater exposure to outdoor aeroallergens, which are elevated locally at those times of year [34]. Interestingly, we found tobacco exposure was positively correlated with total tear IgE as well, even though smoke is not known to cause IgE-mediated sensitization. Tobacco smoke is an irritant associated with decreased desmosome connections in the mammalian conjunctival epithelium [35]. It is plausible that such histopathologic changes result in increased allergen penetration at the ocular surface. This is supported by the fact that exposure exacerbates AC [36], with one study citing tobacco smoke as a trigger for 6.1% of AC patients (*n* = 2817) [8]. Moreover, tobacco smoke has been shown to modulate IgE expression in murine models of allergic disease [37,38].

As with all studies, our findings must be considered in light of the study limitations. First, there are technical concerns in total tear IgE measurement, as sample volumes differed between subjects with variable Schirmer wetting lengths. We attempted to mitigate this limitation by normalizing tear fluid volumes for analysis and then by controlling for tear volume in the multivariable analysis. Indeed, Schirmer length did not significantly associate with IgE, suggesting that his was not a confounding factor in our data. Second, tear IgE levels may be elevated for reasons other than ocular allergy. For example, it is possible that some of our cohort had increased conjunctival vessel permeability secondary to ocular surface inflammation, which could lead to increased total tear protein and IgE content. As such, future studies are needed that evaluate other potential biomarkers of allergy, such as trypase [39], eosinophil cationic protein, interleukins 4, 5, and 13, and matrix metalloproteinase-9 [14], and compare their levels to IgE. It is also possible that applying Schirmer strips contributed to a local inflammatory response that changed the protein milieu of tears. However, we believe that the 5 min duration of exposure would be unlikely to capture a change in protein expression. An alternative tear collection method would have been to use glass capillary tubes [27]. However, this methodology has its own limitation, as it often requires a wash to remove enough tears, especially in individuals with low tear volume or conjunctival chalasis. Third, samples were stored from 6–7 years prior to analysis and it is unknown if IgE was degraded during storage. Fortunately, there was no relationship between storage length and IgE levels. Fourth, the population studied was predominately older male veterans, and DE symptoms and signs were assessed with a set protocol. Future studies in diverse populations and with more robust protocols will be needed to increase generalizability. Finally, we determined exposures by history and not by direct measurements. This is an important avenue of future study, and may be more relevant when measuring allergen-specific tear and serum IgE, because specific exposures in our population were unknown prior to data collection.

Despite these limitations, our study findings suggest that a proportion of individuals with clinical features of DE have an underlying component of ocular allergy. While it is not known if tear IgE levels will be the best biomarker to identify these patients, we found several modifiable exposures associated with total tear IgE levels in a heterogeneous group of individuals without a formal AC diagnosis. Identification of such exposures is crucial for the development of preventative and targeted interventions that are more cost effective and less risky than traditional therapies. Future studies, in diverse and well defined populations (such as those with data on skin testing), will be needed to clarify the role of IgE as a biomarker of allergy in individuals with DE.

## Figures and Tables

**Table 1 jcm-08-01627-t001:** Descriptive statistics of Miami VA Cohort recruited during 2010 and 2011 by IgE status: # of patients (% of patients).

	IgE Level Not Detectable (ND)	Low IgE Level (≤1 ng/mL)	High IgE Level (>1 ng/mL)	*p*-Value
IgE status	18 (24.0%)	44 (58.7%)	13 (17.3%)	
**Race**				0.97
White	12 (24.0%)	29 (58.0%)	9 (18.0%)	
Black	6 (24.0%)	15 (60.0%)	4 (16.0%)	
**Birth place**				0.71
Florida	2 (20.0%)	6 (60.0%)	2 (20.0%)	
US outside FL	8 (20.0%)	24 (60.0%)	8 (20.0%)	
Outside US	7 (36.8%)	9 (47.4%)	3 (15.8%)	
**Psychiatric status**				
Depression symptoms	3 (21.4%)	8 (57.1%)	3 (21.4%)	0.89
PTSD	2 (18.2%)	7 (63.6%)	2 (18.2%)	0.88
**Co-morbidities**				
Hypertension	5 (26.3%)	11 (57.9%)	3 (15.8%)	0.30
Sleep apnea	6 (40.0%)	7 (46.7%)	2 (13.3%)	0.27
Diabetes	6 (28.6%)	12 (57.1%)	3 (14.3%)	0.81
Thyroid,	3 (33.3%)	5 (55.6%)	1 (11.1%)	0.73
Osteoarthritis	10 (30.3%)	18 (54.6%)	5 (15.2%)	0.52
Antihistamine	7 (41.2%)	8 (47.1%)	2 (11.8%)	0.16
**Pet(s) at home**				0.003***
No	14 (32.6%)	26 (60.5%)	3 (7.0%)	
Yes	3 (11.5%)	13 (50.0%)	10 (38.5%)	
**Smoking status**				0.03**
No	18 (25.7%)	42 (60.0%)	10 (14.3%)	
Yes	0 (0.0%)	2 (40.0%)	3 (60.0%)	
**Second-hand smoking exposure at home**				0.02**
No	15 (23.4%)	39 (60.9%)	10 (15.6%)	
Yes	2 (40.0%)	0 (0.0%)	3 (60.0%)	
**Season**				0.03**
Off peak (fall & winter)	14 (31.8%)	26 (59.1%)	4 (9.1%)	
Peak (spring & summer)	4 (12.9%)	18 (58.1%)	9 (29.0%)	
**Environmental allergies**				0.46
No	14 (25.0%)	33 (58.9%)	9 (16.1%)	
Yes	3 (23.1%)	6 (46.2%)	4 (30.8%)	
**Eczema**				0.29
No	17 (27.0%)	34 (54.0%)	12 (19.1%)	
Yes	0 (0.0%)	5 (83.3%)	1 (16.7%)	
**Ventilation by opening windows**				0.11
a week/never	8 (24.2%)	16 (48.5%)	9 (27.3%)	
more than twice/week	10 (23.8%)	28 (66.7%)	4 (9.5%)	

*** *p* < 0.01, ** *p* < 0.05, * *p* < 0.1. PTSD = post-traumatic stress disorder.

**Table 2 jcm-08-01627-t002:** Dry eye parameters of Miami VA Cohort recruited during 2010 and 2011 by IgE status: Mean (standard deviation; # of patients).

Dry Eye Parameter	IgE Status			
	IgE Level Not Detectable (ND)	Low IgE Level (≤1 ng/mL)	High IgE Level (>1 ng/mL)	*p*-Value
**TBUT (s)**	8.7 (1.1; 18)	7.3 (0.7; 44)	9.0 (1.3; 13)	0.37
**Osmolarity (mOsm/L)**	308.9 (2.8; 17)	307.4 (1.7; 44)	311.2 (2.6; 13)	0.81
**Schirmer wetting length (mm/5 min)**	9.9 (1.3; 18)	12.4 (1.0; 44)	13.6 (1.7; 13)	0.23
**Corneal Staining Score**	1.27 (1.01; 18)	1.11 (1.08; 44)	0.69 (0.85; 13)	0.28
**DEQ5 Score**	10.7 (1.3; 18)	11.5 (0.8; 44)	9.4 (1.5; 13)	0.39

TBUT = tear break up time; DEQ5 = Dry Eye Questionnaire-5.

**Table 3 jcm-08-01627-t003:** Association of the selected covariables with IgE levels and IgE categories.

Selected Covariables	IgE Level as Expressed by log_e_ (IgE (ng/mL)) ^a^	IgE Categories ^b^ Odds Ratio
**Age**	0.00992	0.98
	(−0.0375–0.0574)	(0.89–1.09)
**Birth place**	−0.430 *	0.29 **
	(−0.907–0.0476)	(0.11–0.75)
**Smoking status**	1.046 **	38.57 **
	(0.0901–2.002)	(1.90–783.22)
**Tear break up time (second)**	−0.00054	0.93
	(−0.0684–0.0673)	(0.82–1.06)
**Schirmer length (mm)**	−0.032	1.03
	(−0.0996–0.0357)	(0.91–1.17)
**Osmolarity (mOsmol/l)**	−0.00056	1.00
	(−0.0402–0.0391)	(0.93–1.08)
**Allergy status (0 = no, 1 = yes)**	0.366	2.24
	(−0.507–1.238)	(0.51–9.89)
**Eczema status (0 = no, 1 = yes)**	0.187	1.48
	(−0.772–1.146)	(0.25–8.76)
**Pet at home (0 = no, 1 = yes)**	0.777 **	11.54 ***
	(0.00193–1.552)	(2.47–53.79)
**Smoker at home other than the patient (0 = no, 1 = yes)**	1.093 ***	1.44
	(0.351–1.835)	(0.04–57.65)
**Ventilation by opening windows (0 =** **> week or never; 1 = more than once/week)**	−0.29	0.57
	(−0.986–0.406)	(0.19–1.70)
**Season (0 = fall & winter; 1 = otherwise)**	0.38	3.94 **
	(−0.397–1.157)	(1.04–14.91)
**Constant**	−0.503	
	(−13.76–12.76)	
**Observations**	52	68
**R-squared**	0.322	NA

Robust 95% confidence interval in parentheses; *** *p* < 0.01, ** *p* < 0.05, * *p* < 0.1. a = IgE was skewed, therefore, IgE (ng/mL) was fitted at natural log scale. b = IgE was categorized into three groups: 0 = not detected; 1 ≤ 1 ng/mL and 2 > 1 ng/mL; ordinal logistic regression was used to fit the IgE categories. NA = Not applicable.

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
