# Peer review of "Total Tear IgE Levels Correlate with Allergenic and Irritating Environmental Exposures in Individuals with Dry Eye"

_jcm, 2019, doi:10.3390/jcm8101627_

Round 1

Reviewer 1 Report

This paper aims to investigate the presence of markers of allergic inflammation in the tear film of patients of dry eye, and whether their presence correlates with environmental allergen exposure. The authors demonstrate that IgE (total, not allergen specific) is detectable in the tear film of said patients, and high IgE levels correlated with higher levels of allergen/environmental irritant exposure.

Overall the paper is well written, very interesting, and methods have been explained fully with specific detail relating to the selection criteria of patients which is crucial to such studies. However, there are some areas that require attention as described below.

ABSTRACT

line 18: sentence beginning "Total...exposures" is not needed as this is explained later in the abstract anyway.

line 25: the term "Dry Eye Questionnaire 5≥6" appears ambiguous/unclear, please write as "Dry Eye Questionnaire-5 ≥6).

line 26: please avoid starting sentences with a number - "76% had..." - please re-write this sentence.

line 32/33: rather than just stating IgE correlated with allergy season and exposures, please include measure of correlation e.g. p-value.

INTRODUCTION

line 37: the definition of dry eye also symptoms - this should be stated.

line 45: conjunctival papillae are not associated with dry eye, rather it is a sign that differentiates it. Please see (section 9.1.1):

Wolffsohn, J. S., Arita, R., Chalmers, R., Djalilian, A., Dogru, M., Dumbleton, K., ... & Sullivan, B. D. (2017). TFOS DEWS II diagnostic methodology report. The ocular surface, 15(3), 539-574.     EXPERIMENTAL SECTION   Should this section not be called "Methodology"?   line 63: It is not clear why only male subjects were recruited. Also, it is not clear if they were provided with written information prior to consent nor is the form of consent stated (was it verbal, written?).   Given the definition of dry eye including symptoms, why were some subjects with only signs recruited?   This section needs to include the cut-off values used to determine what as defined as dry eye for both symptoms and signs with associated references to demonstrate precedence/validity.   line 89: why were only 2 measures for TBUT taken? - it is advised that 3 are required to obtain reliable average.     RESULTS   line 121: if only male subjects were recruited, why does it state "majority of subjects were male (87%)"? Please explain - it may be that the remaining subjects did not identify as such or wished to provide the information?   DISCUSSION   line 167: The discussion should not start with a concluding statement without providing explanatory rationale. Also, there is spelling mistake in this sentence "with found" should be "we found".   line 173: it may be that the higher amounts of tear IgE are in those with higher tear production simply because there are more tears available to measure IgE at detectable levels - would not a proportion of the IgE per volume tears be clinically useful? Also, the method of tear collection is invasive by nature and may have caused an inflammatory response. Finally, it is not clear of the Schirmer strips were sterile/unexposed to allergens at the time of use.   line 203: I am confused by what measure of tear production (as mentioned in line 173) were used to show higher levels corresponded with higher tear IgE; but here you state that Schirmer wetting length did not associate with IgE? Please can you clarify.                

Author Response

September 18th, 2019

JCM-SI: Dry Eye Syndrome

Re: Total Tear IgE Levels Correlate with Allergenic and Irritating Environmental Exposures in Individuals with Dry Eye

Thank you for considering our manuscript entitled “Total Tear IgE Levels Correlate with Allergenic and Irritating Environmental Exposures in Individuals with Dry Eye” to the Dry Eye Syndrome special issue of JCM. We appreciate the time the reviewers have put into reading our work and have addressed their changes in the point by point below.

ABSTRACT:

line 18: sentence beginning "Total...exposures" is not needed as this is explained later in the abstract anyway. Thank you for the comment. We have now removed the word “total” from the sentence (line 19). It now reads: “Immunoglobulin E (IgE) levels in tear samples were quantified and home environmental exposures assessed via standardized survey.”

line 25: the term "Dry Eye Questionnaire 5≥6" appears ambiguous/unclear, please write as "Dry Eye Questionnaire-5 ≥6). Thank you for giving us the opportunity to clarify. We have added a hyphen to improve clarity between “Questionnaire” and “5” in abstract (line 24-25). Sixty-two individuals had dry eye symptoms (Dry Eye Questionnaire-5≥6) and 75 had one or more signs of DE.”

line 26: please avoid starting sentences with a number - "76% had..." - please re-write this sentence. We have reworded as suggested in the abstract (line 25) “Detectable total tear IgE levels were observed in 76% of subjects, and 17.3% had high levels (>1 ng/mL).”

line 32/33: rather than just stating IgE correlated with allergy season and exposures, please include measure of correlation e.g. p-value. Thank you for the suggestion. We include this information in the abstract (lines 26-29). “Individuals with exposure to pet(s) (odds ratio (OR) 11.5, p=0.002) and smoke (OR 38.6, p=0.008) at home were more likely to have high IgE levels compared to those not exposed. Individuals with tears collected during spring or summer were 3.9 times (p=0.028) more likely to have high IgE compared to those sampled at other times of year.”

INTRODUCTION:

line 37: the definition of dry eye also symptoms - this should be stated. We now include this information in the Introduction (line 38). “Affected individuals present with symptoms of ocular irritation, dryness, discomfort, or foreign body sensation.” (Craig, J. P., et al. (2017). "TFOS DEWS II Definition and Classification Report." Ocul Surf 15(3): 276-283.)

line 45: conjunctival papillae are not associated with dry eye, rather it is a sign that differentiates it. Please see (section 9.1.1): Wolffsohn, J. S., Arita, R., Chalmers, R., Djalilian, A., Dogru, M., Dumbleton, K., ... & Sullivan, B. D. (2017). TFOS DEWS II diagnostic methodology report. The ocular surface15(3), 539-574. Thank you for pointing out this inaccuracy. We have removed “conjunctival papillae” from the sentence (line 46): “However, the two entities have overlapping features that include shared symptoms (discomfort and itching [10]) and signs (bulbar hyperemia and ocular surface inflammation [11-14]).”

 EXPERIMENTAL SECTION:

Should this section not be called "Methodology"?   Thank you for allowing us to clarify. In formatting our work, we referred to the Microsoft word template provided by JCM at: https://www.mdpi.com/journal/jcm/instructions. However, we would be happy to change the section title to “Methodology” if appropriate.

line 63: It is not clear why only male subjects were recruited. Also, it is not clear if they were provided with written information prior to consent nor is the form of consent stated (was it verbal, written?).   We apologize for this error. Both male and female subjects were recruited. This information has been added to the Experimental Section (line 63). “Male and female subjects were prospectively recruited from the Miami VA Medical Center eye clinic between October 2010 and December 2011.” In addition, all subjects provided informed consent. We have revised Experimental Section (line 64) to reflect this. Written informed consent was obtained from all subjects after verbal explanation of the nature and possible consequences of the study.”

Given the definition of dry eye including symptoms, why were some subjects with only signs recruited?   This section needs to include the cut-off values used to determine what as defined as dry eye for both symptoms and signs with associated references to demonstrate precedence/validity.   Thank you for providing the opportunity to clarify our inclusion criteria. The goal of this study was to evaluate the relationship between total tear IgE and dry eye. Given the known disconnect between symptoms and signs of dry eye, we opted to enroll individuals with symptoms and/or signs. We provide information regarding the distribution of symptoms and signs in our population in the results section (line 24) “Sixty-two individuals had dry eye symptoms (DEQ5≥6) and 75 had one or more signs of DE.”

line 89: why were only 2 measures for TBUT taken? - it is advised that 3 are required to obtain reliable average.   Thank you for the opportunity to clarify. We agree that 3 measures generate a more reliable average, however, this project utilized previously collected data that included only two values for TBUT. We have added this as a limitations to the study (lines 223-224). “Fourth, the population studied was predominately older male veterans and DE symptoms and signs were assessed with a set protocol. Future studies in diverse populations and with more robust protocols will be needed to increase generalizability.”

RESULTS:

line 121: if only male subjects were recruited, why does it state "majority of subjects were male (87%)"? Please explain - it may be that the remaining subjects did not identify as such or wished to provide the information?   Thank you for pointing out this discrepancy. We apologize for this error and appreciate the opportunity to correct it. Both male and female subjects were recruited. This information has been added to the Experimental Section as outlined above.

DISCUSSION:

line 167: The discussion should not start with a concluding statement without providing explanatory rationale. Also, there is spelling mistake in this sentence "with found" should be "we found".   Thank you for the recommendation. The first sentence in the Discussion section (line 171) now reads: “We found that a majority of individuals with clinical features of DE but no diagnosis of AC had detectable total tear IgE levels.”

line 173: it may be that the higher amounts of tear IgE are in those with higher tear production simply because there are more tears available to measure IgE at detectable levels - would not a proportion of the IgE per volume tears be clinically useful? We agree that this is an important consideration. We considered two ways to address this. The first was to standardize IgE by volume. We describe this in the Experimental Section (lines 102-105). “Each eluted sample was diluted to a final volume of 200µl using BSS, and equivalent volumes of 100µl of each sample were tested in duplicate. These precautionary volume adjustments were performed to ensure standardization of tear fluid volumes in the samples.” Followed by our rationale “These precautionary volume adjustments were performed to ensure standardization of tear fluid volumes in the samples.” The second was to adjust the coefficients of other covariates for the volume of the tears. We include this information in the results and our rationale for this approach in the discussion (lines 206-209) “We attempted to mitigate this limitation by normalizing tear fluid volumes for analysis and then by controlling for tear volume in the multivariable analysis. Indeed, Schirmer length did not significantly associate with IgE, suggesting that his was not a confounding factor in our data.”

Also, the method of tear collection is invasive by nature and may have caused an inflammatory response. Finally, it is not clear of the Schirmer strips were sterile/unexposed to allergens at the time of use.   Thank you for raising these two important points. We acknowledge that the method of tear collection may have caused a local inflammatory response and that the presence of any foreign material (i.e. the strips themselves) may have changed the protein milieu of tears. However, we believe that the 5-minute duration of exposure would be unlikely to capture a change in protein expression. We add this point to the limitation section (line 214-217) “It is also possible that the test itself contributed to a local inflammatory response that changed the protein milieu of tears. However, we believe that the 5-minute duration of exposure would be unlikely to capture a change in protein expression.” We used sterile Schirmer test strips and have added this information to the Experimental Section (lines 93-94): “Sterile Schirmer test strips were utilized (TearFlo, HUB Pharmaceuticals, Plymouth, MI).”

line 203: I am confused by what measure of tear production (as mentioned in line 173) were used to show higher levels corresponded with higher tear IgE; but here you state that Schirmer wetting length did not associate with IgE? Please can you clarify. Thank you for providing the opportunity to clarify. In our categorical analysis of IgE (stratified into three categories: non-detectable, low and high), mean Schirmer levels were highest in the High IgE group and lowest in the non-detectable group. However, the difference was not statistically significant. This information is provided in Table 2 (lines 144-146). In our multivariable analysis, IgE level (normalized at natural log scale) as a continuous variable was fitted with respect to the selected covariates. In this analysis, Schirmer length did not significantly associate with IgE. We discuss this point on lines 208-209). “Indeed, Schirmer length did not significantly associate with IgE, suggesting that his was not a confounding factor in our data.”

Thank you again for your thoughtful review and for providing us with the opportunity to revise the manuscript.

Thank you for considering our work.

Respectfully,

Anat Galor

Reviewer 2 Report

Page 2

Was oral antihistamine (OTC and Rx) use also part of the exclusion criteria as these agents have anticholinergic properties and

would mimic or exacerbate dye eye?

Page 6

In the first sentence of the discussion, should the word “with” be "we" ?

The criteria of the diagnosis of AC (allergic conjunctivitis) is not maximized by the history of the clinician although does provide a "clinical suspicion" (not necessarily a diagnosis for exclusion). This is a single area of criticism as it would have been improved the actual diagnosis is to have specific serological specific IgE (total IgE is not a specific acceptable criteria for "allergy") but skin testing especially to pets and other perennial allergens as well as trees and grasses which would have correlated to seasonal "spring" release of allergenic pollen. The absence of evidence of peripheral sensitization though negative skin testing and the absence of specific IgE to the Floridian aeroallergen (seasonal an perennial allergens) would strengthen future research into this area.

Irritants are known to act as adjuvants to modulation of the IgE response. Interesting original research by A. Saxon on diesel

exhaust and stimulation of B-cell to induce production of IgE. Heo, Y., A. Saxon and O. Hankinson (2001). "Effect of diesel exhaust particles and their components on the allergen-specific IgE and IgG1 response in mice." Toxicology 159(3): 143-158.

Yang, S. N., C. C. Hsieh, H. F. Kuo, M. S. Lee, M. Y. Huang, C. H. Kuo and C. H. Hung (2014). "The effects of environmental toxins on allergic inflammation." Allergy Asthma Immunol Res 6(6): 478-484.

Nel, A. E., D. Diaz-Sanchez, D. Ng, T. Hiura and A. Saxon (1998). "Enhancement of allergic inflammation by the interaction between diesel exhaust particles and the immune system." J Allergy Clin Immunol 102(4 Pt 1): 539-554.

The other question is that localized IgE production to allergen in tear fluid has been noted in patients with contact lens use of

chymopapain cleansing enzyme. This suggests local production to allergens  for sensitization.

(Bernstein, D. I., J. S. Gallagher, M. Grad and I. L. Bernstein (1984). "Local ocular anaphylaxis to papain enzyme contained in a contact lens cleansing solution." J Allergy Clin Immunol 74(3 Pt 1): 258-260.)

Tobacco smoke induction of IgE production is an interesting point that the authors may include their discussion and for consideration in future studies. Seymour, B. W., J. L. Peake, K. E. Pinkerton, V. P. Kurup and L. J. Gershwin (2005). "Second-hand smoke increases nitric oxide and alters the IgE response in a murine model of allergic aspergillosis." Clin Dev Immunol 12(2): 113-124.; Robbins, C. S., M. A. Pouladi, R. Fattouh, D. E. Dawe, N. Vujicic, C. D. Richards, M. Jordana, M. D. Inman and M. R. Stampfli (2005). "Mainstream cigarette smoke exposure attenuates airway immune inflammatory responses to surrogate and common environmental allergens in mice, despite evidence of increased systemic sensitization." J Immunol 175(5): 2834-2842.

Page 7

The presence of IgE is not necessarily a biomarker of allergy versus IgE production and perhaps sensitization while concomitant measurement of tryptase would have been a better marker of the actual presence of mast cell activation and mediator release into the tear fluid.

Page 10

The questionnaire is based on being told about allergies, but the question from the manuscript is that he patient had skin tests or blood tests suggesting allergy as well that led to the patient’s exclusion.  Where was this data collected from if not form the questionnaire?

Author Response

September 18th, 2019

JCM-SI: Dry Eye Syndrome

Re: Total Tear IgE Levels Correlate with Allergenic and Irritating Environmental Exposures in Individuals with Dry Eye

Thank you for considering our manuscript entitled “Total Tear IgE Levels Correlate with Allergenic and Irritating Environmental Exposures in Individuals with Dry Eye” to the Dry Eye Syndrome special issue of JCM. We appreciate the time the reviewers have put into reading our work and have addressed their changes in the point by point below.

Page 2

Was oral antihistamine (OTC and Rx) use also part of the exclusion criteria as these agents have anticholinergic properties and would mimic or exacerbate dye eye?  Thank you for bringing up this important point. We excluded individuals using topical anti-histamines but not oral anti-histamines. Instead of excluding oral anti-histamines, we assessed this factor with regards to IgE status. Oral anti-histamine use did not significantly associate with IgE status in our study population (p=0.16) (Data provided in Table 1).

Page 6

In the first sentence of the discussion, should the word “with” be "we" ? Thank you for pointing out this mistake. This sentence in the discussion (line 171) now reads: “We found that a majority of individuals with clinical features of DE but no diagnosis of AC had detectable total tear IgE levels.”

The criteria of the diagnosis of AC (allergic conjunctivitis) is not maximized by the history of the clinician although does provide a "clinical suspicion" (not necessarily a diagnosis for exclusion). This is a single area of criticism as it would have been improved the actual diagnosis is to have specific serological specific IgE (total IgE is not a specific acceptable criteria for "allergy") but skin testing especially to pets and other perennial allergens as well as trees and grasses which would have correlated to seasonal "spring" release of allergenic pollen. The absence of evidence of peripheral sensitization though negative skin testing and the absence of specific IgE to the Floridian aeroallergen (seasonal an perennial allergens) would strengthen future research into this area. Thank you for raising this important point. We agree that future research into this area would be strengthened by studying a population with negative skin testing and absence of specific IgE to commonly encountered Floridian aeroallergens. We add this point to the limitations section (lines 233-235). “Future studies, in diverse and well defined populations (such as those with data on skin testing), will be needed to clarify the role of IgE as a biomarker of allergy in individuals with DE.“

Irritants are known to act as adjuvants to modulation of the IgE response. Interesting original research by A. Saxon on diesel exhaust and stimulation of B-cell to induce production of IgE: Heo, Y., A. Saxon and O. Hankinson (2001). "Effect of diesel exhaust particles and their components on the allergen-specific IgE and IgG1 response in mice." Toxicology 159(3): 143-158. Yang, S. N., C. C. Hsieh, H. F. Kuo, M. S. Lee, M. Y. Huang, C. H. Kuo and C. H. Hung (2014). "The effects of environmental toxins on allergic inflammation." Allergy Asthma Immunol Res 6(6): 478-484. Nel, A. E., D. Diaz-Sanchez, D. Ng, T. Hiura and A. Saxon (1998). "Enhancement of allergic inflammation by the interaction between diesel exhaust particles and the immune system." J Allergy Clin Immunol 102(4 Pt 1): 539-554. Thank you for bringing this to our attention. While we do not address diesel exhaust directly, we acknowledge the potential for irritants to modulate IgE expression in the discussion (lines 202-203). “Moreover, tobacco smoke has been shown to modulate IgE expression in murine models of allergic disease.[37, 38]”

The other question is that localized IgE production to allergen in tear fluid has been noted in patients with contact lens use of chymopapain cleansing enzyme. This suggests local production to allergens for sensitization. (Bernstein, D. I., J. S. Gallagher, M. Grad and I. L. Bernstein (1984). "Local ocular anaphylaxis to papain enzyme contained in a contact lens cleansing solution." J Allergy Clin Immunol 74(3 Pt 1): 258-260.) Thank you for bringing this interesting point to our attention. We excluded patients with a history of contact lens wear from the study. Experimental section (line 70-71). “patients were excluded from participation if they had ocular or systemic conditions that could confound DE, including contact lens wear...”

Tobacco smoke induction of IgE production is an interesting point that the authors may include their discussion and for consideration in future studies. Seymour, B. W., J. L. Peake, K. E. Pinkerton, V. P. Kurup and L. J. Gershwin (2005). "Second-hand smoke increases nitric oxide and alters the IgE response in a murine model of allergic aspergillosis." Clin Dev Immunol 12(2): 113-124.; Robbins, C. S., M. A. Pouladi, R. Fattouh, D. E. Dawe, N. Vujicic, C. D. Richards, M. Jordana, M. D. Inman and M. R. Stampfli (2005). "Mainstream cigarette smoke exposure attenuates airway immune inflammatory responses to surrogate and common environmental allergens in mice, despite evidence of increased systemic sensitization." J Immunol 175(5): 2834-2842. Thank you for providing this information and agree that this is an important point to consider, addressed in the discussion as noted above.

Page 7

The presence of IgE is not necessarily a biomarker of allergy versus IgE production and perhaps sensitization while concomitant measurement of tryptase would have been a better marker of the actual presence of mast cell activation and mediator release into the tear fluid. Thank you for bringing this important point to our attention. We agree that there is a need to study other potential biomarkers for allergy and have added this to the discussion (lines 212-214). “As such, future studies are needed that evaluate other potential biomarkers of allergy, such as trypase[39], eosinophil cationic protein, interleukins 4, 5, and 13, and matrix metalloproteinase-9 [14], and compare their levels to IgE.”

Page 10

The questionnaire is based on being told about allergies, but the question from the manuscript is that the patient had skin tests or blood tests suggesting allergy as well that led to the patient’s exclusion.  Where was this data collected from if not form the questionnaire? Thank you for raising this point. This information came from review of the medical records. This information is now provided in the Experimental section (lines 69-70). “Based on review of the medical record, patients were excluded from participation…”

Thank you again for your thoughtful review and for providing us with the opportunity to revise the manuscript.

Thank you for considering our work.

Respectfully,

Anat Galor